# Effects of Warm-Up on Sprint Swimming Performance, Rating of Perceived Exertion, and Blood Lactate Concentration: A Systematic Review

**DOI:** 10.3390/jfmk6040085

**Published:** 2021-10-19

**Authors:** Olivia Czelusniak, Emily Favreau, Stephen J. Ives

**Affiliations:** Health and Human Physiological Sciences Department, Skidmore College, Saratoga Springs, NY 12866, USA; oczelusniak99@gmail.com (O.C.); eefavs@gmail.com (E.F.)

**Keywords:** warm up, swimming, performance, rating of perceived exertion, blood lactate

## Abstract

(1) Background: warm-ups precede physical exertion and has been shown to have positive and negative effects on performance. Positive effects include elevating body temperature, heart rate, and VO_2_. Negative effects include increasing fatigue and blood lactate concentration. The most effective warm-up format is still unknown, particularly in competitive swimming. The purpose of this systematic review was to determine the most beneficial warm-up for maximal performance in sprint swimming events; (2) Methods: a structured search was carried out following the Preferred Reporting Items for Systematic review and Meta-Analyses (PRISMA) guidelines in the PubMed, SportDiscus, and Google Scholar databases until March 2021. Studies with double-blind and randomized designs in which different types of warm-up were compared to each other or an identical placebo condition (no warm-up) were considered. Fourteen published studies were included. The effects of warm-up on sprint swimming performance, rating of perceived exertion (RPE) and blood lactate concentration (La^−^) were investigated. (3) Results: in half of the studies, swimmers performed significantly better after a regular warm-up; however, the effect of warm-up on performance was small. Warm-ups had a medium to large effect on RPE and a small to medium effect on La^−^. (4) Conclusions: the findings of this review suggest that warm-ups do influence performance, although the magnitude is small. Future studies are needed in larger populations to clarify whether warm ups improve swim performance, to what extent, and the potential role of variables related to participant characteristics and swimming competitions.

## 1. Introduction

A warm-up is commonly defined as any physical activity performed before a main event, such as practice or competition. Common warm-ups include cardio activities like calisthenics, dynamic stretching, or sport-specific movements. For example, baseball players may practice their swing, and track athletes may practice their starts from the racing blocks. 

The positive effects of warm-ups have been well-documented [1,2,3,4,5]. A warm-up has been proven to increase body and core temperature and elevate baseline VO_2_, which can contribute to improved physical performance [2,6]. Another benefit is decreased risk of injury. As heart rate and body temperature increase, blood flow increases, which facilitates the delivery of oxygen to muscles, decreasing stiffness and preparing muscles for more strenuous exercise [2]. Although the positive effects of warm-up have been well-documented, negative effects have also been noted. Long warm-ups have been reported to cause fatigue and may detract from performance [2,7]. Different sports also utilize different muscles and energy systems, thus each requiring a specialized warm-up that accounts for the unique nature of each sport to maximize performance [2]. Therefore, identifying the most optimal warm-up for a sport is critical. 

These issues are all present in the current research about the role of a warm-up in competitive swimming. Swimming is considered an intermediate to long-term activity. Any activity with a duration of more than 10 s but less than 5 min is considered an intermediate activity, while activities 5 min or longer are considered long-term [2]. A warm-up for swimming is important, since it decreases muscle stiffness, positively alters the force-velocity relationship, increases baseline VO_2_ and reduces the risk of injury [8,9,10]. In competition, a warm-up allows swimmers to adjust to the conditions of the competition pool and better prepare themselves for their races [11,12,13].

In competitive swimming, there are many types of warm-ups [8,11,14,15,16]. Active warm-ups, in which the athlete physically moves around, may be performed in the water, on land (known as a dryland warm-up), or in both environments. Passive warm-ups, in which a swimmer remains seated and wears heat-conserving clothing, are also utilized. However, there is a lack of consensus about which is the most beneficial for maximizing performance. It is speculated that most warm-up procedures are based on trial and error rather than scientific evidence [2,13]. Previous research on different types of warm-up have not determined clear effects. Small sample sizes, poor controls, and lack of proper statistical analyses all contribute to the unclear results reported on warm-ups [2,5]. Many studies have examined the physiological effects of warm-ups, but have failed to determine the effects of a warm-up on performance [3,8,15,16,17,18]. 

Thus, there is a need to identify the most beneficial warm-up for maximal performance in competitive swimming. The purpose of this systematic review was to determine the current most beneficial warmup for maximal performance in sprint swimming events. 

## 2. Materials and Methods

### 2.1. Searching Strategies

The present article is a systematic review focusing on the effects of different types of warm-up on sprint swimming performance. It was carried out following the preferred reporting items for systematic review and meta-analyses (PRISMA) guidelines, which helped to improve the integrity of this review [19]. The PICOS model was used to determine the inclusion criteria: P (population): swimmers; I (intervention): active or passive warm-up; C (comparators): same conditions with control or placebo; O (outcome): swimming trial time and effects on rating of perceived exertion (RPE) and blood lactate (La^−^); and S (study design): single-blind and randomized design. 

A structured search was conducted in the following databases: PubMed, SportDiscus, and Google Scholar. It included results until 30 March 2021, with searches restricted to articles published in the past 20 years (Figure 1). Search terms were related to warm-up and swimming performance. The following terms were used: “warm up” (all fields) AND “swimming performance” (all fields]. All titles and abstracts from the search were cross-referenced to identify duplicates and any potential missing studies. Titles and abstracts were screened for a subsequent full-text review. Both authors (OC and EF) performed a search for relevant studies and any disagreements were resolved through discussion between them. 

### 2.2. Inclusion and Exclusion Criteria

No restrictions were applied to the swimmers’ level, age, or sex to increase the range of studies eligible for analysis. However, there were several inclusion criteria used to select studies from the articles obtained in the database search. These were studies: [1] in which the only difference between the experimental and control conditions was the type of warm-up performed; [2] testing the effects of different types of warm-up on sprint swimming performance; [3] with a randomized design; [4] conducted as investigational trials, not meta-analyses or systematic reviews; and [5] published in English. The following exclusion criteria were applied to the experimental protocols of the review: [1] studies that were not conducted with swimmers; [2] studies that did not specifically examine the effect of a warm-up on sprint swimming performance; and [3] studies that were published beyond the last 20 years.

### 2.3. Data Extraction

Once the inclusion/exclusion criteria were applied to each study, data on each study source (including authors and year of publication), study design, participant characteristics (gender and level), sample size, type of warm-up, and final outcomes of the interventions were extracted independently by the authors using a spreadsheet. Any disagreements were resolved through discussion between them. 

All studies included measurements of swimming performance from time trials. They also all included measurements of two or more types of performance outcomes (e.g., rating of perceived exertion and blood lactate level). Eight studies included measurements of RPE and five studies included measurements of blood lactate level. 

### 2.4. Statistical Analysis

The mean (M), standard deviation (SD), and sample size data were extracted by the authors from the tables of all the included papers. Descriptive data of the participants’ characteristics are reported as mean ± standard deviation. Descriptive analyses were performed using a spreadsheet (Microsoft Excel 2016, Redmond, WA USA). In seven studies, fixed effect sizes of the performance outcomes analyzed were calculated using an online effect size calculator calculating Cohen’s d (Social Science Statistics, London, UK). In the other seven studies, effect sizes were extracted from the calculations provided in the articles [7,12,17,20]. The Cohen criteria were used to interpret the magnitude of the effect sizes: 0.2–0.5, small; 0.5–0.8, moderate; and >0.8, large. 

## 3. Results

### 3.1. Main Search

The literature search identified 25 relevant studies. However, only 14 articles met all the inclusion criteria (see Figure 1). From the 25 articles, there were no duplicates, so none were removed at this stage. Two articles were excluded because they were not full-text and were not available to the researchers. From the 23 full-text articles assessed for eligibility, another 9 papers were removed because they did not meet all the inclusion criteria. The topics and numbers of studies excluded were: 6 articles that were systematic reviews, not clinical trials; 1 review article that did not discuss the effects of warm-up on swimming performance; 1 article on the effects of warm-up on other sports, such as running and cycling; and 1 article that focused on recovery time after warm-up, not warm-up itself. Thus, the current systematic review included 14 studies. 

**Figure 1 jfmk-06-00085-f001:**
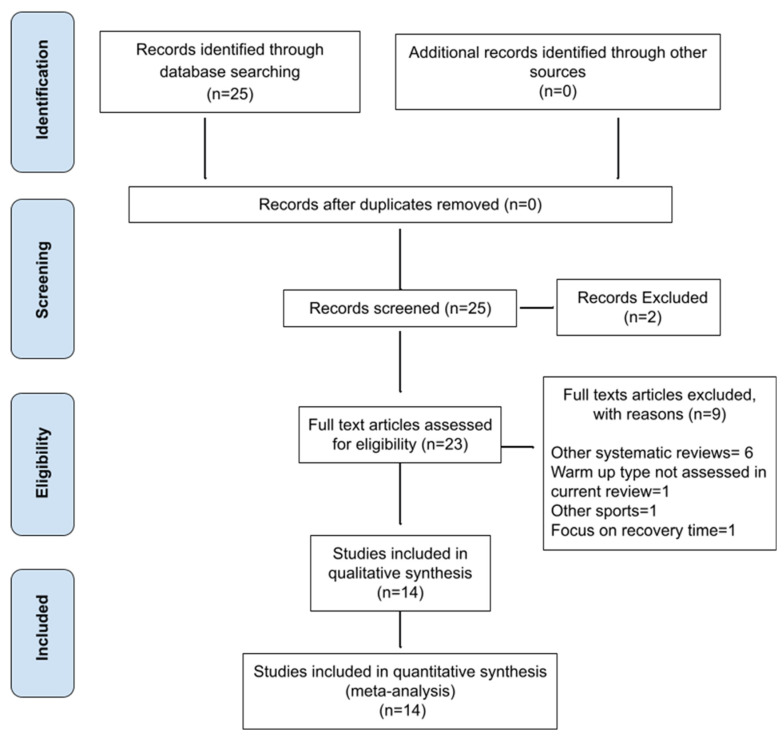
Selection of studies.

### 3.2. Testing of Various Warm-Up Procedures

The participant and intervention characteristics of the studies included in this systematic review are depicted in Table 1, whereas the summary of studies included is shown in Table 2. The total sample consisted of 204 swimmers (116 males, 88 females). The most common age of participants was “college-age”.

Table 1 shows the samples included in all studies (“n” refers to the number of studies). The samples consisted of swimmers who competed at national, elite, and sub-elite levels (n = 6), in NCAA Division I programs (n = 3), at a competitive level (n = 4), and at an age-group level (n = 1). The ages of participants ranged from college-age (n = 9) to teenagers (14–17 years old, n = 4) to young adults (15–25 years old, n = 1). In six studies, the effects of active warm-ups were tested, while in one study, the effects of passive warm-ups were tested. The remaining seven studies tested the effects of both active and passive warm-ups. Most studies tested the effects of swimming warm-ups performed in the water (n = 6), but others tested the effects of dryland activities (n = 1) or warm clothing (n = 3). Other studies tested the effects of multiple types of warm-up strategies (n = 4). For in-water warm-ups, the length that swimmers completed was not uniform. In the studies that were analyzed, the shortest warm-up that swimmers completed was 50 yards [8], while the longest warm-up that swimmers completed was 3000 m [3]. In other studies, swimmers completed warm-ups that were longer than 50 yards but shorter than 3000 m. The length of the time trial swimming races that swimmers completed were more consistent. In two studies, a 50-yard distance was used, while 50 m was used in four studies and 100 m was used in eight studies. In most studies, swimmers performed freestyle in the time trial (n = 12). In other studies, swimmers performed the breaststroke (n = 1) or freestyle and breaststroke (n = 1).

Table 2 summarizes all studies included in the systematic review and meta-analysis. It shows the participant characteristics, interventions, whether rating of perceived exertion and blood lactate concentration were recorded in the study, and any significant effects of warm-up on RPE and La^−^ that were recorded. Several studies (n = 8) supported the theory that a warm-up has a positive effect on sprint swimming performance, while the others (n = 6) did not support this theory, noting that a warm-up did not influence sprint swimming performance. A warm-up was shown to increase RPE in four studies and increase blood lactate concentration in two studies. 

### 3.3. Effect of Warm-Up on Time Trial Performance, Rating of Perceived Exertion, and Blood Lactate

Table 3 shows the effect sizes calculated for time trial performance (TTP), rating of perceived exertion (RPE), and blood lactate concentration (La^−^). TTP was measured in all studies, while RPE was reported in ten studies and La^−^ was reported in 5 studies. A warm-up was shown to have a small effect on TTP in most studies (n = 11). For RPE, a warm-up had a medium effect in most studies (n = 5). The effect of a warm-up on blood lactate concentration was less uniform. It had a small effect in two studies, a medium effect in two studies, and a large effect in one study. 

Several studies compared multiple warm-ups to each other, and thus, there were more values for effect sizes [3,8,11,12,21,22]. The most complex study was by Kafkas et al. [12]. In the study, a warm-up had varying effects on TTP. In the trials where swimmers performed the freestyle stroke, a warm-up had mostly small effects on TTP, but some medium and large effects were recorded. In the trials where a breaststroke was performed, a warm-up had mostly small effects on TTP. In terms of RPE, large effects were seen in the pre-trial measurements, while all small effects were recorded post-trial.

## 4. Discussion

The purpose of this systematic review was to evaluate the effects of warm-up on sprint swimming performance and determine the most effective method. The results were inconclusive. It was found that the type of warm-up (active, passive, a combination of the two, etc.) had a small effect on time trial performance. Yet, in several studies, it was found that after a standard active warm-up, swimmers performed significantly better [4,11,12,21]. However, in other studies, there was no significant difference in performance after completing different types of warm-up [3,8,15,16,17,18]. There was a medium to large effect on RPE and a small to medium effect on blood lactate concentration. The results, though contradictory, indicate there is a relationship between warm-up and swimming performance. The exact nature of that relationship remains to be determined. 

### 4.1. Impact of Warm-Up on Sprint Swimming Performance

In half the studies (n = 7), it was concluded that swimmers performed significantly better after a regular warm-up than over a shorter or dryland warm-up [4,6,7,11,12,21,22]. However, a warm-up had a small effect on time trial performance across most studies [3,6,7,8,11,15,16,17,18,20,22]. This suggests there is some discontinuity among and within studies, but also suggests that there is a relationship between a warm-up and swimming performance. A warm-up impacts the body’s physiology and primes the athlete to perform at a high intensity with a lower risk of injury [2,6]. Understanding the specific impact a warm-up has on time would benefit both coaches and swimmers. 

### 4.2. Impact of Warm-Up on Rating of Perceived Exertion

A warm-up had a medium to large effect on RPE after time trials in most of the studies [7,11,16,17,18,20]. This finding demonstrates that, overall, RPE is affected by a warm-up. The body after a warm-up is looser, and more likely to perform at maximum levels [2,6]. RPE during a time trial or any race is important for performance. If a swimmer’s perceived exertion is lower after a warm-up than after no warm-up or little warm-up, that swimmer will be able to push themselves harder and thus perform better. This is important for coaches as they begin to plan their warm-ups, and for swimmers as they warm up.

### 4.3. Impact of Warm-Up on Blood Lactate Concentration

A warm-up had a small to medium effect on blood lactate concentration [4,16,18,20,21]. This indicates a relationship between warm-up and performance. Two studies that measured lactate saw a significant increase in lactate after the time trial [20,21], meaning the swimmers were able to reach a higher exercise threshold. The ability to get to a higher threshold is important for peak performance. If a swimmer does not perform the right amount of warm-up, their ability to reach the lactate threshold will be impacted, as the metabolic pathways will not be primed or will become overworked [2]. This will lead to decreases in performance. 

The combination of time trial performance, RPE, and lactate results as stated above are most relevant for swimmers and coaches. A warm-up affects the body’s physiology, decreasing muscle stiffness, elevating baseline VO_2_, and decreasing oxygen debt, which elevates aerobic capacity. This decrease in muscle stiffness and elevation in aerobic capacity impacts RPE [2,6]. This impact can cause significant improvements in performance as the swimmer feels better and is able to perform better. The elevation of blood lactate reveals that the swimmer has performed at a high intensity [2]. Research indicates that warming up at a severe intensity will cause the athlete to reach metabolic acidemia, which has been seen to impair subsequent performance [2]. Warming up too little would impact the ability to reach lactate thresholds [2]. Being able to warm up at the right intensity primes the body to reach the elevated lactate levels that are associated with high intensity performance without impairing that performance [2]. As coaches prepare their warm-ups, they should consider how different warm-ups will affect performance, and should structure these warm-ups in ways that are most likely to benefit their swimmers. Similarly, swimmers should consider the impact of their warm-up on their performance, particularly when they are warming up without a coach. Utilizing this knowledge may encourage them to practice more effective warm-ups that may help maximize their performance. 

### 4.4. Limitations

The main limitations of this meta-analysis are that not all studies researched were included and that risk of bias was not calculated. Many studies were excluded because they were incomplete, meaning they did not have the full text. Future research should include more studies and should calculate risk of bias.

## 5. Conclusions

The findings of this meta-analysis suggest that a warm-up influences performance. To what extent performance is influenced is still undetermined. Performance time was significantly impacted by a warm-up, but the effect size of a warm-up was small. There was a medium to large effect on RPE and a small to medium effect on blood lactate. Future studies should look more closely at recovery periods after a warm-up, such as rest during dual swim meets versus championships, or rest between warm-up and time trial. Future studies should also examine the differences in swimmers’ age, sex differences, and/or differences between sprinters and distance swimmers. More complex investigations of this topic will greatly benefit both coaches and swimmers.

## Figures and Tables

**Table 1 jfmk-06-00085-t001:** Participant and intervention characteristics of the studies included in the systematic review.

Level of participants	National level Elite Sub-elite Competitive NCAA Division I Age-group	4 studies [16,17,21,22] 1 study [20] 1 study [12] 4 studies [4,7,8,18] 3 studies [8,11,15] 1 study [6]
Age group of participants	College-age 14-17 years 15-25 years	9 studies [3,7,8,11,12,15,17,20,22] 4 studies [4,6,16,18] 1 study [21]
Type of warm-up	Active Passive Active & passive	6 studies [3,6,8,15,18,21] 1 study [7] 7 studies [4,11,12,16,17,20,22]
Specific warm-up used	In-water Dry-land Warm clothing Multiple types	6 studies [3,4,11,16,18,21] 1 study [22] 3 studies [7,17,20] 4 studies [6,8,12,15]
Length of warm-up	1100 yards & 50 yards 100 yards & swimmer’s choice 3000 m & 1500 m 2150 m 2000 m 1200 m 1000 m 1200, 600, & 1800 m	1 study [8] 1 study [11] 1 study [3] 1 study [15] 1 study [6] 2 studies [12,18] 2 studies [4,16] 1 study [21]
Time trial distance	50 yards 50 m 100 m	2 studies [8,11] 4 studies [12,15,16,22] 8 studies [3,4,6,7,17,18,20,21]
Time trial stroke	Freestyle Breaststroke Freestyle & breaststroke	12 studies [3,4,6,7,8,11,15,16,18,20,21,22] 1 study [17] 1 study [12]

**Table 2 jfmk-06-00085-t002:** Summary of studies included in the systematic review.

Author/s-Year	Population	Intervention ^	Were RPE and La^−^ Recorded?	Time Trial Results	Significant Effects on RPE and La^−^
Al-Nawaiseh et al. (2012) [8]	13 college-age Division I swimmers (4 f, 9 m)	Swim (1100 yd), combo (swim & plyometrics), short (50 yd swim)	No	WU did not significantly influence times	N/A
Balilionis et al. (2012) [11]	16 college-age Division I swimmers (8 m, 8 f)	None, short (100 yd swim), regular (swimmer’s normal WU)	Yes, RPE	Significantly faster times after regular vs. short WU	↑ RPE after regular WU, no significant effect on La^−^
Dalamitros et al. (2018) [22]	19 national swimmers, ages 17–23 (10 m, 9 f)	During transition phase: power (exercise circuit), stretch (dynamic/active stretching), or passive WU	Yes, RPE	Males: significantly faster times after power WU; females: faster times after stretch WU	No significant effects reported
Dimitrić et al. (2012) [3]	12 swimmers, age 19–26 (8 m, 4 f)	Swim WU: long (3000 m), short (1500 m), or high-intensity (1500 m at an intense pace)	No	WU length did not significantly influence times	N/A
Galbraith & Willmott (2018) [7]	9 college-age swimmers (3 f, 6 m)	Passive: warm (T-shirt, hooded top, pants, gloves, socks, sneakers) or limited clothing (T-shirt)	Yes, RPE post-sprint	Significantly faster times (0.6%) in warm condition	No significant effects on RPE or La^−^ reported
Kafkas et al. (2019) [12]	14 sub-elite college-age female swimmers	Without stretch (passive rest), static stretch (10 min), in-water (1200 m), dryland (10 exercises)	Yes, RPE	Significantly faster times after in-water WU	↑ RPE after in-water WU, no significant effects on La^−^
McGowan et al. (2017) [20]	25 college-age elite swimmers (12 m, 13 f)	1350 m swim; then transition using: heated jacket + dryland (combo) or regular jacket + seated (control)	Yes, La^−^ and RPE	Times were significantly faster (0.8%) after combo WU	↑ RPE after combo, ↑ La^−^ after control WU vs. combo WU
McGowan et al. (2016) [17]	10 college-age national swimmers (6 m, 4 f)	Same WU as in study [12]	Yes, RPE	WU did not significantly influence times	↑ RPE after combo WU, no significant effects on La^−^
Moran (2012) [15]	16 Division I college-age swimmers (5 f, 11 m)	Static stretch (9 min) or dynamic WU (9 exercises), & 2150 m swim	No	WU did not significantly influence times	N/A
Neiva et al. (2014) [4]	20 competitive swimmers, ages 15–17 (10 m, 10 f)	With WU (1000 m swim) or without WU	Yes, RPE and La^−^	Significantly faster times in WU condition	No significant effects reported
Neiva et al. (2016) [18]	13 competitive male swimmers, ages 15–20	700 m swim, then 4 × 25 m race pace swim-control or 8 × 50 m aerobic swim-experimental	Yes, RPE and La^−^	WU did not significantly influence times	↑ RPE after time trial with exp. WU, no effects on La^−^
Neiva et al. (2015) [21]	11 male national swimmers, ages 15–25	Standard (1200 m swim), short (600 m swim), long (1800 m swim)	Yes, RPE and La^−^	Significantly faster times after standard and short WUs	↑ La^−^ after standard and short WU, no significant effects on RPE
Neiva et al. (2012) [16]	7 female national swimmers, ages 14–16	With WU (1000 m swim) or without WU	Yes, RPE and La^−^	WU did not significantly influence times	No significant effects reported
Thomas & Goodwin (2013) [6]	19 age group swimmers, ages 12–19 (12 m, 7 f)	2000 m swim + 40 min rest or 2000 m swim + dryland	No	Significantly faster times after swim + dryland	N/A

↑: statistically significant increase; WU: warm-up; RPE: rating of perceived exertion; La^−^: blood lactate concentration; ^ swim WUs were primarily performed using freestyle, but many included a mix of kicking, pulling, and stroke drills as well.

**Table 3 jfmk-06-00085-t003:** Effect sizes calculated for time trial performance (TTP), rating of perceived exertion (RPE), and blood lactate concentration (La^−^).

Author/s-Year	Time Trial Performance (TTP)	Rating of Perceived Exertion (RPE)	Blood Lactate (La^−^)
Al-Nawaiseh et al. (2012) [8]	Short vs. swim: 0.02 Short vs. combo: 0.05 Swim vs. combo: 0.07	None reported	None reported
Balilionis et al. (2012) [11]	None vs. short: 0.04 None vs. regular: 0.15 Short vs. regular: 0.19	None vs. short: 0.05 None vs. regular.: 0.43 Short vs. regular.: 0.58	None reported
Dalamitros et al. (2018) [22]	Females: Power vs. stretch: 0.49 Power vs. control: 0.31 Stretch vs. control: 0.21 Males: Power vs. stretch: 0.29 Power vs. control: 0.445 Stretch vs. control: 0.08	Reported, no significant differences (reported as “2–3” in both power and stretch)	None reported
Dimitrić et al. (2012) [3]	Long vs. short: 0.01 Long vs. high-intensity: 0.09 Short vs. high-intensity: 0.08	None reported	None reported
Galbraith & Willmott (2018) [7]	0.1	0.6	None reported
Kafkas et al. (2019) [12]	Freestyle: WS vs. SS: 0.47 WS vs. IW: 0.50 WS vs. DL: 0.25 SS vs. IW: 0.94 SS vs. DL: 0.75 IW vs. DL: 0.28 Breaststroke: WS vs. SS:0.28 WS vs IW: 0.64 WS vs DL: 0.28 SS vs IW: 0.47 SS vs. DL: 0.20 IW vs. DL: 0.24	Freestyle: WS vs. SS: 0.13 WS vs. IW: 0.04 WS vs. DL: 0.23 SS vs. IW: 0.17 SS vs. DL: 0.35 IW vs. DL: 0.18 Breaststroke: WS vs. SS: 0.25 WS vs IW: 0.05 WS vs DL: 0.16 SS vs IW: 0.21 SS vs. DL: 0.42 IW vs. DL: 0.22	None reported
McGowan et al. (2017) [20]	0.21	0.77	−1.29
McGowan et al. (2016) [17]	−0.05	0.51	None reported
Moran (2012) [15]	0.03	None reported	None reported
Neiva et al. (2014) [4]	0.69	0.41	0.32
Neiva et al. (2016) [18]	0.07	0.82	0.56
Neiva et al. (2015) [21]	WU vs short WU: 0.09 WU vs. long WU: 0.95 Short WU vs. long WU: 1.12	WU vs short WU: 0.09 WU vs. long WU: 0.24 Short WU vs. long WU: 0.17	WU vs short WU: 0.68 WU vs. long WU: 0.69 Short WU vs. long WU: 0.25
Neiva et al. (2012) [16]	0.15	0.62	0.41
Thomas & Goodwin (2013) [6]	0.07	None reported	None reported

WU: warm-up; WS: without stretch; SS: static stretch; IW: in-water; DL: dryland.

## Data Availability

Not applicable.

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
