# Peer review of "Effects of Warm-Up on Sprint Swimming Performance, Rating of Perceived Exertion, and Blood Lactate Concentration: A Systematic Review"

_jfmk, 2021, doi:10.3390/jfmk6040085_

Round 1

Reviewer 1 Report

General comments

The present work attempt to address the question of the effectiveness of warm up on sprint swimming performance, though a systematic review and meta-analysis methodology.

Although the research work done by the authors in the literature seems to be promising, to my perspective more elaboration is needed to the presentation of their findings.

The authors present a short description of the problem in the introduction. Some more details were needed to give focus to the current evidence before the following analysis. The analysis itself although methodologically seems correct it does not adequately support the first part of the purpose, the review of literature. The presentation is rather descriptive and does not follow an in depth analysis and critical perspective of the findings.  The purpose of obtaining an answer in a research question is not facilitated adequately due to this lack of detailed information. Regarding the meta- analysis, the statistical process of the gathered data from studies that measured similar variables is not evident. The tables shown or the written text do not represent any results of meta-analysis. As a general comment regarding the meta-analysis the authors must keep in mind that the purpose of the methodology is to improve the estimation of the result, as it is an overall result (global view of the studied subject). In order to succeed in that end, an appropriate model should be chosen and subsequently present and critically discuss the results. This is not evident in the manuscript. Furthermore, a selection of an appropriate metric on which the meta-analysis will be based is of paramount importance. For example, if the data are continuous in nature such metrics could be mean difference or standardized mean difference or in the case of binary data metrics like odds ratio, risk ratio and risk difference should be adopted. Another important issue that needs clarification is the type of meta-analysis performed. Meta-analysis usually performed with fixed effects of random effects. Which is the case here? Some experts argue that, since clinical and methodological diversity always occur in a meta-analysis, statistical heterogeneity is inevitable (Higgins 2003). Thus, the test for heterogeneity is irrelevant to the choice of analysis and should always be performed. This not evident in the manuscript. To conclude more effort should be done by the authors at the following points.

A detailed analysis of the literature with critical comments.

A clear data analysis of similar studies (i.e. comparing same variables between different protocols) with summarized statistical results.  

A discussion must follow based on their statistical results from this meta-analysis in comparison with the first section of their study.

Specific comments

Abstract

Line 9: The authors could include more details concerning representative parameters as positive and negative effects.

Introduction

Lines 43-44: Authors should consider that this sentence is from the abstract of the article that is cited. It could be rephrased.

Lines 50-51: In this sentence authors could correct their text compilation as it is hard for the reader to understand the essence of this sentence.

Line 53: Authors could use “which of the above mentioned” instead of “these”.

Line 54: Authors should take consideration that while what is stated in a scientific study can be cited, it does not mean that it is 100% valid. I recommend the term “It is speculated that”

before the word  “most” .

Line 59: Authors could use a number of different original scientific studies in which is cited that there is no clear effects of warm up in performance rather than citing text from a review article.

Material and methods

Lines 87-89: Regarding the inclusion criteria it is not clear what difference no 1. has compared to the others. More details are needed.

Line 100: Authors could use “gender” instead of “sex”.

Results

Line 138: The average and std values of the selected studies supplementary to the “14 to 26” years could be used.

Line 140: Authors could note more clearly that n is referred to studies.

Line 144: Add the word “both” after tested.

Line 147-148: Add “that” between “length swimmers” and “that” after warm ups. In general until the end of the paragraph English editing is necessary.

Line 166: Authors could include the age of the participants and the stroke in the population and intervention category. In the same table more information is needed. For example regarding stretching was it active or passive? (Dalamitros et al. 2018) also in the next paper high intensity is referred to short long or both, in Nieva et al. 2015 what does standard warm up represents? Also at the previous study what is “Tcorenet”? In general there are a lot abbreviations in this table, which although some of them are reported they make reading difficult.

Line189: table 3.

Authors could find a uniform structure to cite the TTP effect size on this column. 

If they choose to cite the gender in one of the articles cited in the column then they should cite it on every study.The same applies for other variables too, ex: Freestyle. Also thy should find a more consistent way to report note in more detail the effect size and the warm up that is reported in those studies

McGowan et al. (2016)

Moran (2012)

Neiva et al. (2014)

Neiva et al. (2017)

Neiva  et al. (2012)

Thomas & Goodwin (2013)

Discussion

Line 195: Perhaps use the term blood lactate concentration.

4.1, 4.2 & probably 4.3 that is missing as a heading:

Authors should give detailed results of some of the studies included in the discussion section so the reader could better understand the effect of statistical differences that warm-up has on different variables.

Line 231: Authors should use citations of the original articles than from the results stated in a review. Even if the review is well structured and in great detail there might be differences in the written language used or even mistakes. Authors should validate the citations with the original studies throughout the whole article.The term acidemia in this study is cited differently in the review article (5) (section 2.4 in the review article of Bishop D.J.) as acidaemia

Lines 235-237: Authors should rephrase this sentence -As coaches ….. performance-. The meaning of the sentence doesn’t add up to a clear understanding and conclusion and might confuse the readers. Moreover as it is well stated in this review and in other studies it is not clear that the impact of the warm-up on the performance is only positive.

Author Response

COMMENT: The present work attempt to address the question of the effectiveness of warm up on sprint swimming performance, though a systematic review and meta-analysis methodology. Although the research work done by the authors in the literature seems to be promising, to my perspective more elaboration is needed to the presentation of their findings.

RESPONSE: We appreciate the constructive feedback of the reviewer, the general enthusiasm, and for their time in reviewing the current manuscript. The feedback received has been instrumental in helping us reimagine the manuscript.

COMMENT: The authors present a short description of the problem in the introduction. Some more details were needed to give focus to the current evidence before the following analysis. The analysis itself although methodologically seems correct it does not adequately support the first part of the purpose, the review of literature. The presentation is rather descriptive and does not follow an in-depth analysis and critical perspective of the findings.  The purpose of obtaining an answer in a research question is not facilitated adequately due to this lack of detailed information. Regarding the meta- analysis, the statistical process of the gathered data from studies that measured similar variables is not evident. The tables shown or the written text do not represent any results of meta-analysis. As a general comment regarding the meta-analysis the authors must keep in mind that the purpose of the methodology is to improve the estimation of the result, as it is an overall result (global view of the studied subject). In order to succeed in that end, an appropriate model should be chosen and subsequently present and critically discuss the results. This is not evident in the manuscript. Furthermore, a selection of an appropriate metric on which the meta-analysis will be based is of paramount importance. For example, if the data are continuous in nature such metrics could be mean difference or standardized mean difference or in the case of binary data metrics like odds ratio, risk ratio and risk difference should be adopted. Another important issue that needs clarification is the type of meta-analysis performed. Meta-analysis usually performed with fixed effects of random effects. Which is the case here? Some experts argue that, since clinical and methodological diversity always occur in a meta-analysis, statistical heterogeneity is inevitable (Higgins 2003). Thus, the test for heterogeneity is irrelevant to the choice of analysis and should always be performed. This not evident in the manuscript. To conclude more effort should be done by the authors at the following points.

A detailed analysis of the literature with critical comments.

A clear data analysis of similar studies (i.e. comparing same variables between different protocols) with summarized statistical results.  

A discussion must follow based on their statistical results from this meta-analysis in comparison with the first section of their study.

RESPONSE: We have scaled back the paper to focus on the review, as we felt in is not within our purview to address the meta-analysis with the proper treatment it deserves, but this review provides an excellent foundation for others to undertake such a task. Thank you for these comments and helping us arrive at this decision.

Specific comments

COMMENT: Abstract Line 9: The authors could include more details concerning representative parameters as positive and negative effects.

RESPONSE: Increased heart rate has been included as a positive effect of warm-up, and increased blood lactate concentration has been included as a negative effect of warm-up.

COMMENT: Introduction Lines 43-44: Authors should consider that this sentence is from the abstract of the article that is cited. It could be rephrased.

 RESPONSE: Thank you for this note. The sentence has been rephrased.

COMMENT: Lines 50-51: In this sentence authors could correct their text compilation as it is hard for the reader to understand the essence of this sentence.

 RESPONSE: This sentence has been edited for clarity.

COMMENT: Line 53: Authors could use “which of the above mentioned” instead of “these”.

 RESPONSE: Thank you, this edit has been made.

COMMENT: Line 54: Authors should take consideration that while what is stated in a scientific study can be cited, it does not mean that it is 100% valid. I recommend the term “It is speculated that” before the word “most”.

RESPONSE: Agreed, changed as proposed.

COMMENT: Line 59: Authors could use a number of different original scientific studies in which is cited that there is no clear effects of warm up in performance rather than citing text from a review article.

RESPONSE: The proposed edit has been made, and six original studies are cited rather than the previous citation from a review article.

COMMENT: Material and methods Lines 87-89: Regarding the inclusion criteria it is not clear what difference no 1. has compared to the others. More details are needed.

 RESPONSE: Inclusion criteria number 1 has been clarified.

COMMENT: Line 100: Authors could use “gender” instead of “sex”.

 RESPONSE: The proposed edit has been made.

COMMENT: Results Line 138: The average and std values of the selected studies supplementary to the “14 to 26” years could be used.

RESPONSE: The raw data from all studies was not available, so it was not possible to calculate the average and standard deviation of the participants’ ages. The “14 to 26 years” detail was removed.

COMMENT: Line 140: Authors could note more clearly that n is referred to studies.

RESPONSE: The proposed edit has been made.

COMMENT: Line 144: Add the word “both” after tested.

 RESPONSE: The proposed edit has been made.

COMMENT: Line 147-148: Add “that” between “length swimmers” and “that” after warm ups. In general until the end of the paragraph English editing is necessary.

 RESPONSE: The proposed edit has been made, and the paragraph has been edited for clarity.

COMMENT: Line 166: Authors could include the age of the participants and the stroke in the population and intervention category. In the same table more information is needed. For example regarding stretching was it active or passive? (Dalamitros et al. 2018) also in the next paper high intensity is referred to short long or both, in Nieva et al. 2015 what does standard warm up represents? Also at the previous study what is “Tcorenet”? In general there are a lot abbreviations in this table, which although some of them are reported they make reading difficult.

 RESPONSE: The age of the participants has been included in the table. A note about the stroke used during warm-up has been included at the end of the table. The note indicates that during swimming warm-ups, swimmers primarily swam freestyle, but performed kicking, pulling, and stroke drills as well. More information has been added about the studies mentioned, and the table has been edited for clarity, removing most of the confusing abbreviations. 

COMMENT: Line189: table 3.Authors could find a uniform structure to cite the TTP effect size on this column. 

RESPONSE: The table has been modified.

COMMENT: If they choose to cite the gender in one of the articles cited in the column then they should cite it on every study. The same applies for other variables too, ex: Freestyle. Also thy should find a more consistent way to report note in more detail the effect size and the warm up that is reported in those studies

McGowan et al. (2016)

Moran (2012)

Neiva et al. (2014)

Neiva et al. (2017)

Neiva  et al. (2012)

Thomas & Goodwin (2013)

 RESPONSE: Gender was specifically examined in Dalamitros et al. (2018), so the effect sizes were calculated by the authors according to how each gender performed after completing each type of warm-up. Effect sizes were not calculated according to gender in the other studies. Similarly, in the study by Kafkas et al. (2019), two sets of time trials were conducted, one in which the swimmers swam freestyle and the other in which swimmers swam breaststroke. As such, the effect sizes were calculated according to these methods. In all other trials, the swimmers swam freestyle.

COMMENT: Discussion Line 195: Perhaps use the term blood lactate concentration.

 RESPONSE: The proposed edit has been made.

COMMENT: 4.1, 4.2 & probably 4.3 that is missing as a heading:

RESPONSE: The heading for section 4.3 has been clarified and the heading for the Limitation section has been added (now section 4.4).

COMMENT: Authors should give detailed results of some of the studies included in the discussion section so the reader could better understand the effect of statistical differences that warm-up has on different variables.

 RESPONSE: Given this is a focused review we will not be focusing on these details.

COMMENT: Line 231: Authors should use citations of the original articles than from the results stated in a review. Even if the review is well structured and in great detail there might be differences in the written language used or even mistakes. Authors should validate the citations with the original studies throughout the whole article.The term acidemia in this study is cited differently in the review article (5) (section 2.4 in the review article of Bishop D.J.) as acidaemia

 RESPONSE: We appreciate the suggestion but as stated by the reviewer the review article does nicely summarize the relevant articles.  As for acidemia vs. acidaemia that is a British-English vs. International-English phenomenon which often simply coincides with a journal being based in England. As the current journal does not require the “ae” we would use the more international spelling.

COMMENT: Lines 235-237: Authors should rephrase this sentence -As coaches ….. performance-. The meaning of the sentence doesn’t add up to a clear understanding and conclusion and might confuse the readers. Moreover as it is well stated in this review and in other studies it is not clear that the impact of the warm-up on the performance is only positive.

RESPONSE: Thank you for this suggestion. The lines have been rephrased to make the meaning clearer. The sentence and those that follow now indicate that warm-up may improve performance, but this is not assured.

Reviewer 2 Report

Hello,

the Article need to include also information about warm up content details. The tables are too long. In discussion are no information about design quality of presented studies. I miss also the health and injurise aspect of warm up procedures. The Name of the Article don´t correspond with content, where are not information only about perfomance but also about lactate...

Author Response

COMMENT: Hello,

the Article need to include also information about warm up content details. The tables are too long. In discussion are no information about design quality of presented studies. I miss also the health and injurise aspect of warm up procedures. The Name of the Article don´t correspond with content, where are not information only about perfomance but also about lactate...

RESPONSE: We appreciate the constructive feedback of the reviewer, the general enthusiasm, and for their time in reviewing the current manuscript. We have made significant edits to the manuscript, including those raised above. We believe the manuscript is now much improved, thank you.

Round 2

Reviewer 2 Report

no comments